# AN EMPIRICAL STUDY ON NORMALIZATION IN MAMBA

## ABSTRACT

Normalization layers are crucial for improving the training efficiency and stability of deep neural network architectures. The recently proposed Mamba network has demonstrated significant potential in competing with Transformers. However, as with many deep architectures, the training stability of Mamba remains a significant challenge, and normalization techniques are key to addressing this issue. In this paper, we systematically investigate the effects of normalization type, position and combinations on the Mamba Block. On the one hand, we conducted extensive experiments to evaluate the impact of applying various normalization layers before or after the SSM module(the core module of Mamba Block). On the other hand, we performed thorough experiments to assess the effects of combining diverse normalization techniques before and after the SSM module. Our analysis encompasses both long sequence modeling and image classification tasks. The results show that applying normalization layers after the SSM module (if used only once) and combining different normalization layers before and after the SSM module can enhance training stability and improve Mamba performance. Furthermore, we provide practical recommendations for selecting appropriate normalization techniques in designing Mamba architectures and validated them on other datasets. We hope that our insights will help mitigate training instabilities in deep learning and foster the development of more robust architectures. All codes and models used in this study will be open-sourced on GitHub.

## 1 INTRODUCTION

The Mamba architecture is a state-of-the-art model designed to efficiently handle long sequences in sequence modeling by leveraging selective structured state space models (Selective SSM) (Gu et al., 2021). However, training Mamba presents significant challenges due to the inherent unpredictability of SSM (Hamilton, 1994). Normalization plays a crucial role in effectively training deep neural networks (Huang et al., 2023), which provides a potential solution for training Mamba. There are various usages of normalization techniques in the variants of Mamba architecture (Gu & Dao, 2023; Bai et al., 2024b; Liu et al., 2024; Gong et al., 2024; Ting et al., 2024), but without providing the sufficient evidence for supporting why one should use like that. Are there any patterns in the use of normalization in Mamba? And how can we better utilize normalization techniques in Mamba? To answer these questions, it is necessary to conduct a comprehensive investigation into various normalization strategies. In this study, we systematically evaluate the effect of **normalization types, positions, and combinations** on Mamba's performance and stability in sequence modeling and image classification tasks. We also provide valuable recommendations on how to select normalization techniques when designing new Mamba frameworks.

For **normalization types**, different normalization methods have distinct characteristics, functions, and applicable scenarios. Mamba and its variants have utilized diverse normalization methods (Gu & Dao, 2023; Bai et al., 2024b; Liu et al., 2024; Gong et al., 2024; Ting et al., 2024). However, no work has yet explored the impact of these normalization methods on Mamba's performance or provided general recommendations on choosing a suitable normalization technique. To address this, we explore the impact of several widely used normalization techniques, including BN, LN, GN, IN, and RMSN, in both sequence modeling and image classification tasks.

For **normalization positions**, using normalization layers at different positions can adjust the data distribution at those locations, thereby affecting the model's performance. Currently, many Mamba variants utilize normalization layers at different positions relative to the SSM module, such as (Ma et al., 2024; Chen et al., 2024d; Huang et al., 2024; Liu et al., 2024; Chen et al., 2024e). This raises a question: what is the optimal position to apply normalization? To address this question, we explored the impact of applying normalization layers before or after SSM on Mamba's performance.

For **normalization combinations**, Different normalization methods adjust data distributions from various perspectives. Currently, several Mamba variants adopt different normalization methods before and after the SSM. On one hand, some variants use the same normalization method both before and after SSM, such as (Zhou et al., 2024; Chen & Ge, 2024; Bai et al., 2024b; Gong et al., 2024). On the other hand, some variants use different normalization methods before and after SSM, respectively, such as (Fan et al., 2024; Dong et al., 2024a; Hatamizadeh & Kautz, 2024; Qu et al., 2024; Ju & Zhou, 2024;?). Can combining different normalization methods achieve better performance? What is the optimal combination strategy? These questions remain unanswered. Therefore, it is necessary to explore the impact of applying different combinations of normalizations on Mamba's performance. To this end, we investigated the effects of pairwise combinations among five common normalization methods.

We analyzed the results of the above three exploratory experiments. Our analysis, rooted in the scale invariance of deep learning (Papyan, 2018), revealed that the weight matrices (Huang et al., 2020) in deeper Mamba blocks exhibited significantly larger L2 norms (Luo et al., 2016) than those in earlier layers. Introducing normalization before the SSM provides almost no help with this issue, whereas placing normalization after the SSM helped the model maintain consistency in the scale of the L2-norms across different layers during training and even making the weight updates more stable, thereby improving training stability. Moreover, applying an appropriate combination of normalization before and after the SSM further helps address this issue, making the model training more stable and leading to better performance.

We also validated our analysis conclusions on other datasets. We selected the optimal Mamba normalization scheme and conducted experiments on other sequence modeling and image classification tasks. Compared with the original benchmarks, the experimental results demonstrated that our proposed optimization scheme achieved better performance.

**Contributions** In this work, we systematically investigate various mainstream normalization techniques and aim to address the following questions:

(1) Normalization Type and Position: What is the optimal applicable normalization method? Where should normalization be placed in the model to ensure high performance? We found that using different normalization methods impacts the performance significantly, and applying normalization after SSM generally leads to better model performance than applying it before SSM.

(2) Normalization Combination: What are the effects of normalization combinations? And what is the intuition behind these effects? We found that certain combinations of different normalization techniques produce more excellent results. A view of the L2 norm of the weight matrix in the Mamba Block can be the institution of this phenomenon.

(3) Combination intuition: How can we choose the proper combination? We propose an intuition for harmonizing normalization strategies in deep architectures, providing practical guidelines for selecting normalization methods. We also validated our findings on other datasets.

## 2 RELATED WORK

In the Mamba architecture, the SSM module is the core component, and its performance is highly sensitive to the choice of normalization strategies. Normalization adjusts feature distributions to improve model stability and generalization. Thus, studying the types of normalization, their positioning relative to the SSM module, and the impact of combining different normalization methods on performance is crucial. Based on this, we categorize the related works into four groups: **no normalization, normalization before SSM, normalization after SSM, and combined normalization both before and after SSM**, as shown in Figure 1.

**No Normalization:** In this group, many Mamba variants directly optimize the model structure or design task-specific solutions without employing normalization techniques. For example, in sequence tasks, models such as XLSR-Mamba (Xiao & Das, 2024b), SpikMamba (Chen et al., 2024c), UmambaTSF (Wu et al., 2024c), SWIM (Zhang et al., 2024b), TF-Mamba (Xiao & Das, 2024a), MetaMamba (Kim, 2024), PackMamba (Xu et al., 2024), MaTrRec (Zhang et al., 2024a), TransMA (Wu et al., 2024b), and ECGMamba (Qiang et al., 2024) have demonstrated good performance by refining module structures or parameter-sharing mechanisms. In vision tasks, models like I2I-Mamba (Atli et al., 2024) and DeMamba (Chen et al., 2024a) avoid normalization by designing task-driven modules. These works focus on exploring the intrinsic potential of the Mamba architecture but may face instability in handling complex tasks or multimodal inputs.

**Normalization before SSM:** This strategy aims to stabilize feature distributions before entering the SSM module, thereby enhancing its robustness to input features. In sequence tasks, various nor-

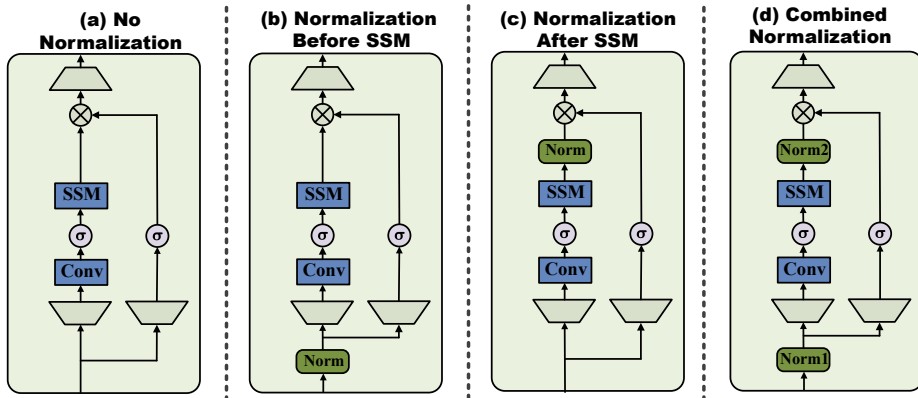

Figure 1: Four categories of related normalization works on Mamba.

malization methods are widely used. For instance, LN is employed in FMamba (Ma et al., 2024), MLSA4Rec (Su & Huang, 2024), and Zamba (Glorioso et al., 2024), while RMSN is applied in DiMSUM (Phung et al., 2024), Quamba (Chiang et al., 2024), bi-CrossMamba (Wu et al., 2024a), Mamba-PTQ (Pierro & Abreu, 2024), and CMAMBA (Zeng et al., 2024). BN and IN are also used in self-supervised (Liang et al., 2024) and MC-SEMamba (Ting et al., 2024), respectively. In vision tasks, LN is the most commonly adopted method, appearing in models such as RetinexMamba (Bai et al., 2024a), CU-Mamba (Deng & Gu, 2024), and RSMamba (Chen et al., 2024d), while RMSN is used in FST-Mamba (Wei et al., 2024) to improve feature stability. This body of work demonstrates that applying normalization before SSM effectively mitigates input feature distribution shifts, providing stable inputs for the computations within the SSM module.

**Normalization after SSM:** Unlike the previous approach, this strategy focuses on adjusting feature distributions after the SSM module to optimize subsequent processing. In sequence tasks, RMSN is used in DIFFIMP (Gao et al., 2024), IN is adopted by Bi-Mamba (Tang et al., 2024), and GN is applied in Mamba2 (Bai et al., 2024b). In vision tasks, LN is employed in IRSRMamba (Huang et al., 2024), while GN is used in MambaHSI (Li et al., 2024). The advantage of this strategy lies in its ability to fine-tune the feature distributions processed by the SSM module, aligning them with the requirements of downstream tasks and significantly enhancing model performance in some complex scenarios.

**Combined Normalization both before and after SSM:** This strategy applies normalization at both positions to balance feature distributions and optimize overall performance. In sequence tasks, some models, such as MambaDC (Chen et al., 2024e), Bi-Mamba (Zhou et al., 2024), and TiM4Rec (Fan et al., 2024), apply LN both before and after SSM. Others use different combinations: BMAMBA2 (Bai et al., 2024b) applies GN at both positions, DepMamba (Fan et al., 2024) uses BN before SSM and LN after SSM, while SSD4Rec (Qu et al., 2024) applies LN before SSM and GN after SSM. In vision tasks, similar combined normalization methods are widely applied. For instance, models such as ChangeMamba (Chen et al., 2024b), MiM-ISTD (Chen et al., 2024f), and Fusion-Mamba (Dong et al., 2024b) apply LN both before and after SSM. Hamba (Dong et al., 2024a) and MambaVision (Hatamizadeh & Kautz, 2024) use BN before SSM and LN after SSM, while MSS (Ju & Zhou, 2024) and VM-DDPM (Ju & Zhou, 2024) adopt GN before SSM and LN after SSM. This group demonstrates significant performance improvements, although the optimal configuration may vary depending on the task and data distribution.

In conclusion, applying normalization in the Mamba architecture exhibits diversity and significance. Normalization before SSM effectively stabilizes inputs, making it suitable for handling complex features. Normalization after SSM is more effective for optimizing downstream task performance. Combined normalization methods adjust feature distributions both before and after SSM, offering more opportunities for performance enhancement. However, choosing normalization methods and combination strategies still requires further exploration tailored to specific tasks. Therefore, it is necessary to conduct a systematic investigation into the normalization methods used in Mamba, which could provide valuable directions and references for future Mamba framework designs.

# 3 METHOD

## 3.1 THE FORMATION OF MAMBA BLOCK

Mamba is a state-of-the-art architecture based on the discrete SSM (State Space Model). The overall structures of the Mamba Block or its variations are shown in Figure 2. The processes of the Mamba Block can be described as follows:

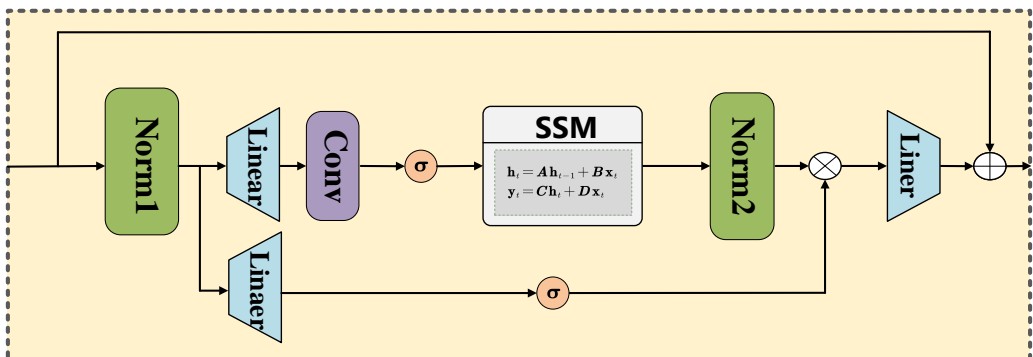

Figure 2: The structure of Mamba Block or its variations.

First, the input $x$ is normalized using $N1$, followed by a series of transformations in the main branch:

$$f = N2(SSM(Act(Con(Lin(N1(x)))))). \tag{1}$$

Where $N1$ is the first Normalization, $Lin$ is the Linear projection, $Con$ is the Depth-Wise convolution, $Act$ is the SiLU activation, $SSM$ is the selective structured state space models (selective SSM, SSM), $N2$ is the Second normalization. Meanwhile, in parallel, the normalized input $N1(x)$ is passed through another Linear projection and SiLU activation:

$$p = Act(Lin(N1(x))). \tag{2}$$

Then, the main and parallel branches' outputs are combined via element-wise multiplication $u = f \odot p$. Finally, the combined result is passed through a linear projection, and the original input $x$ is added as a residual connection:

$$y = Lin(f \odot p) \oplus x. \tag{3}$$

Mamba introduces a selective mechanism that allows the parameters of SSM to be dynamically adjusted based on the input tokens. In addition, it improves the SSM computation method, achieving sub-quadratic computational complexity. The following is a brief review of SSM.

For sequence modeling, let the sequence define as $X = \{\mathbf{x}_1, \mathbf{x}_2, \ldots, \mathbf{x}_T\} \in \mathbb{R}^{T \times D}$, where $T$ represents the sequence length and $D$ denotes the number of feature channels. Each element $\mathbf{x}_t = \{x_t^1, x_t^2, \ldots, x_t^D\} \in \mathbb{R}^D$ contains $D$ feature values at time step $t$. For a mini-batch of data with $m$ samples, the input tensor is typically shaped as $(m, D, T)$. The state space model(SSM) is commonly known as a Linear Time-Invariant (LTI) system that transfers the input activation $\mathbf{x}_t \in \mathbb{R}^D$ into output $\mathbf{y}_t$ through the hidden state $\mathbf{h}_t \in \mathbb{R}^H$, which can be denoted as:

$$\mathbf{y}_t = SSM_{\boldsymbol{A},\boldsymbol{B},\boldsymbol{C},\boldsymbol{D}}(\mathbf{x}_t). \tag{4}$$

Its internal computations are defined as follows:

$$\mathbf{h}_t = \boldsymbol{A}\mathbf{h}_{t-1} + \boldsymbol{B}\mathbf{x}_t, \quad \mathbf{y}_t = \boldsymbol{C}\mathbf{h}_t + \boldsymbol{D}\mathbf{x}_t, \tag{5}$$

where $A \in \mathbb{R}^{H \times H}$, $B \in \mathbb{R}^{H \times D}$, $C \in \mathbb{R}^{D \times H}$, $D \in \mathbb{R}^{D \times D}$ are learnable parameters of SSM, $H$ is the number of hidden feature channels.

Apart from handling one-dimensional sequences, it can also effectively process two-dimensional (2D) image data. Specifically, the image is firstly flattened to convert the shape of $h \times w$ into a

sequence of length $T$ before being input into the model. Our work primarily focuses on studying the impact of the type, position, and combination of $N1$ and $N2$ normalization on the performance of Mamba. Next, we will provide a detailed explanation of the main modules.

## 3.2 NORMALIZATION TYPES

Normalization is a critical technique in deep learning aimed at stabilizing and accelerating the training process of neural networks (Desjardins et al., 2015). It adjusts the input data distribution or intermediate activations to a normal distribution, facilitating the convergence of model training (Huang et al., 2023). By incorporating normalization layers, models can maintain consistent representation learning and effectively handle varying input scales. Therefore, selecting the appropriate normalization method according to specific requirements is essential for training models with complex data or large parameter sizes. The common forms of normalization are defined as follows:

$$\text{Norm}(x) = \frac{x - E[x]}{\sqrt{Var[x] + \epsilon}} \cdot \gamma + \beta, \tag{6}$$

Where $E[x]$ represents the mean of the input $x$, $\text{Var}[x]$ represents the variance of the input $x$, and $\epsilon$ is a small constant added to prevent division by zero. $\gamma$ and $\beta$ are the scale and shift parameter, respectively. Many normalization methods have been proposed, but the following five are still commonly used: BN, LN, GN, IN, and RMSN. They are suitable for most models and the most widely adopted methods by researchers. For more information, please refer to appendix A.

## 3.3 NORMALIZATION POSITIONS

The position of normalization layers relative to the SSM module significantly impacts the performance and stability of the Mamba architecture. These positions can be categorized into two cases: before SSM or after SSM.

**Normalization Before SSM** Normalization layers placed before SSM stabilize the input features by reducing shifts in their distribution, which can be formulated as:

$$f = SSM(Act(Con(Lin(N1(x))))), \tag{7}$$

Where $N1$ represents the normalization applied before the SSM module. This positioning ensures SSM operates on well-conditioned inputs, minimizing potential disruptions caused by unstable feature distributions.

**Normalization After SSM** Normalization layers placed after SSM refine the feature distribution generated by SSM. This can be expressed as:

$$f = N2(SSM(Act(Con(Lin(x))))), \tag{8}$$

Where $N2$ represents the normalization applied after the SSM module. This configuration aligns processed features with the requirements of downstream tasks, thereby enhancing performance.

## 3.4 NORMALIZATION COMBINATIONS

Normalization combinations involve using different normalization techniques at different positions around the SSM module. By leveraging the strengths of various methods, this approach adjusts feature distributions more effectively, enhancing stability and performance. Combining normalizations before and after SSM can be expressed as:

$$f = N2(SSM(Act(Con(Lin(N1(x)))))). \tag{9}$$

We examine two main combination strategies: same normalization at both positions and different normalization at each position. $N1 = N2$ represents the same normalization method applied at both positions, maintaining uniformity feature adjustments and avoiding conflicts between normalization methods. For instance: $N1 = N2 = LN$ in VMamba, $N1 = N2 = GN$ in BMAMBA2. $N1 \neq N2$ represents different normalizations applied at each position, taking advantage of different techniques to optimize feature processing at various stages. For example: $N1 = BN$, $N2 = LN$ in DepMamba,

stabilizing inputs and refining outputs. $N1 = \text{LN}, N2 = \text{GN}$ in SSD4Rec, balancing input stability and output performance.

Our research aims to identify the optimal normalization settings by adjusting the type, position, and combination of normalization techniques. The objective function for the Mamba can be expressed as the minimization of the loss between the output and the ground truth label. The complete formulation is as follows:

$$\mathcal{L} = \min_{\Theta} \mathcal{L}_{\text{loss}}\big(\text{Lin}(f \odot p) \oplus x, G\big), \tag{10}$$

where the $f$ and $p$ are defined in The Formation of Mamba Block section, $G$ is the ground truth label. This formulation defines the optimization problem for training the Mamba, incorporating its detailed computation process and the ground truth target $G$. The parameter $\Theta$ includes all learnable components: the weights and biases of $Lin, Con, SSM$, and the scale and shift parameters $(\gamma, \beta)$ of $N1$ and $N2$. By optimizing the objective function, it is possible to mitigate the limitations of individual methods, improve parameter updates, and enhance convergence stability. These findings provide a practical guideline for designing robust normalization strategies in the Mamba architecture.

## 4 EXPERIMENTAL

This section presents a series of systematic experiments to evaluate the impact of normalization types, positions, and combination strategies on the performance and stability of the Mamba Block. We assessed the model's performance on two types of tasks: long sequence modeling and vision tasks. Subsequently, we analyzed and interpreted the experimental results based on the scale invariance of deep learning and the distribution characteristics of the weight norm iterations.

### 4.1 DATASETS

For long sequence modeling, we use the Breakfast dataset (Kuehne et al.). Breakfast is a large-scale dataset designed for evaluating models on long sequence modeling and activity segmentation. It consists of 1,712 videos recorded in 18 different kitchens, involving 52 participants performing 10 distinct actions related to breakfast preparation, such as making tea, frying eggs, and preparing toast. Each video is annotated with frame-level action labels, with sequences often comprising multiple nested actions. The dataset spans over four million frames, making it highly challenging for models to handle long temporal dependencies effectively.

For vision tasks, we use the ImageNet-100 dataset (Krizhevsky et al., 2012). ImageNet-100 is a randomly selected subset of the ImageNet-1k dataset from the 2012 Large Scale Visual Recognition Challenge. It contains 100 categories, covering various objects and scenes, ensuring diversity in vision tasks. The training set includes 1300 images per category, while the validation set contains 50 images per category, totaling 135,000 images.

We also evaluate our proposed optimized normalization scheme on ImageNet-1k (Krizhevsky et al., 2012) and LRA ListOps dataset (Tay et al., 2021) to compare it with other methods.

### 4.2 IMPACT OF DIFFERENT NORMALIZATION TYPES

The performance of different normalization types applied both before and after the SSM module is summarized in the following Table 1. For instance, None→SSM→None indicates that no normalization is applied either before or after the SSM module. BN→SSM→BN indicates that BN is applied both before and after the SSM module, and others are similar in meaning.

Table 1: Performance with Different Normalization Types.

| Normalization Method | Sequence Accuracy (%) | Image Accuracy1 (%) |
|---|---|---|
| None→SSM→None | 7.0 | 10.7 |
| BN→SSM→BN | 41.4 | 74.6 |
| IN→SSM→IN | 40.6 | 83.7 |
| LN→SSM→LN | **58.9** | **86.6** |
| RMSN→SSM→RMSN | 56.9 | 84.1 |
| GN→SSM→GN | **68.8** | **86.3** |

In the **sequence modeling task**, applying GN before and after SSM improved performance from the baseline of 7.0% (no normalization) to 68.8%. **LN** and **RMSN** also significantly enhanced

performance to 58.9%, making it the next-best performer. Other normalization methods like IN and BN provided moderate improvements.

In the **image classification task**, applying LN before and after SSM boosted accuracy to 86.6%, a substantial increase from the 10.7% baseline. **GN** closely followed with an accuracy of 86.3%, demonstrating its strong performance. RMSN and IN also improved accuracy significantly.

These results indicate that the choice of normalization method significantly affects performance and that **GN shows consistently strong performance across both tasks**, achieving high accuracy and making it a reliable choice when aiming for balanced performance. However, the optimal normalization method differs between tasks: GN excels in sequence modeling, likely due to its ability to stabilize training with sequential data, while LN are better suited for image classification, possibly because they normalize across features in a way that benefits spatial data.

### 4.3 Effect of Normalization Positions Relative to SSM

To facilitate the comparison of normalization positions within each task, we integrated the results of normalization before SSM and after SSM into one table as follows: Table 2 and Table 3.

Table 2: Sequence Modeling Performance with Normalization Applied Before or After SSM

| Normalization Method | Accuracy (%) | Normalization Method | Accuracy (%) |
|---|---|---|---|
| None→SSM→None | 7.0 | None→SSM→None | 7.0 |
| BN→SSM→None | 28.4 | None→SSM→BN | 28.4 |
| IN→SSM→None | 10.9 | None→SSM→IN | 7.0 |
| LN→SSM→None | 57.1 | None→SSM→LN | 59.1 |
| RMSN→SSM→None | 58.7 | None→SSM→RMSN | 60.5 |
| GN→SSM→None | 20.5 | None→SSM→GN | **70.1** |

Table 3: Image Classification Performance with Normalization Applied Before or After SSM

| Normalization Method | Accuracy1(%) | Normalization Method | Accuracy1 (%) |
|---|---|---|---|
| None→SSM→None | 10.7 | None→SSM→None | 10.7 |
| BN→SSM→None | 20.5 | None→SSM→BN | 67.8 |
| IN→SSM→None | 70.2 | None→SSM→IN | 83.8 |
| LN→SSM→None | 86.5 | None→SSM→LN | **86.7** |
| RMSN→SSM→None | 86.3 | None→SSM→RMSN | 84.2 |
| GN→SSM→None | 66.1 | None→SSM→GN | **86.8** |

In **sequence modeling**, applying GN after SSM improved performance to 70.1%, compared to 20.5% when GN was applied before SSM. LN and RMSN showed consistent performance, with a slight advantage when applied before SSM (57.1% and 58.7%) versus after SSM (59.1% and 60.5%). Overall, After-SSM tends to yield better or comparable results except for IN.

For **image classification**, applying GN after SSM achieved the highest accuracy of 86.8%, significantly outperforming GN before SSM (66.1%). Similarly, LN after SSM attained an accuracy of 86.7%, slightly higher than LN before SSM (86.5%). These results suggest that after-SSM is more effective for image classification as well except for RMSN.

In most cases, **applying normalization after SSM** is more beneficial than **applying normalization before SSM**. This suggests that normalizing the output of SSM is crucial for enhancing performance. In sequence modeling and image classification, applying GN after SSM provides substantial performance gains.

### 4.4 Effects of Combining Different Normalization Methods

The performances of different normalization combinations in both sequence modeling and image classification are in the following Table 4. Their bar charts are shown in Figure 3.

In **sequence modeling**, the combination of IN before SSM and LN after SSM led to the highest performance of 72.5%, indicating that different normalization methods can complement each other. IN also showed strong performance when combined with RMSN after SSM (72.2%), reinforcing its versatility.

For **image classification**, the highest accuracy of 87.3% was achieved with RMSN before SSM and BN after SSM. Combinations involving BN after SSM generally resulted in high accuracies,

Table 4: Sequence Modeling and Image Classification Performance with Different Normalization Combinations

| Normalization Method | Sequence Accuracy (%) | Image Accuracy1 (%) |
|---|---|---|
| None→SSM→None | 7.0 | 10.7 |
| BN→SSM→BN | 41.4 | 74.6 |
| BN→SSM→IN | 63.1 | 83.2 |
| BN→SSM→LN | 56.5 | 86.3 |
| BN→SSM→GN | 70.1 | 86.1 |
| BN→SSM→RMSN | 57.8 | 84.6 |
| IN→SSM→BN | 51.3 | 86.7 |
| IN→SSM→IN | 67.6 | 83.7 |
| IN→SSM→LN | **72.5** | 85.7 |
| IN→SSM→GN | 70.1 | 85.3 |
| IN→SSM→RMSN | 72.2 | 83.5 |
| LN→SSM→BN | 52.1 | 87.1 |
| LN→SSM→IN | 66.8 | 84.5 |
| LN→SSM→LN | 58.9 | 86.6 |
| LN→SSM→GN | 69.3 | 86.6 |
| LN→SSM→RMSN | 57.4 | 84.6 |
| GN→SSM→BN | 52.1 | 87.1 |
| GN→SSM→IN | 63.4 | 84.5 |
| GN→SSM→LN | 71.9 | 86.3 |
| GN→SSM→GN | 68.8 | 86.3 |
| GN→SSM→RMSN | 68.1 | 68.1 |
| RMSN→SSM→BN | 41.4 | **87.3** |
| RMSN→SSM→IN | 71.4 | 84.3 |
| RMSN→SSM→LN | 56.5 | 86.1 |
| RMSN→SSM→GN | 70.7 | 85.7 |
| RMSN→SSM→RMSN | 56.9 | 84.1 |

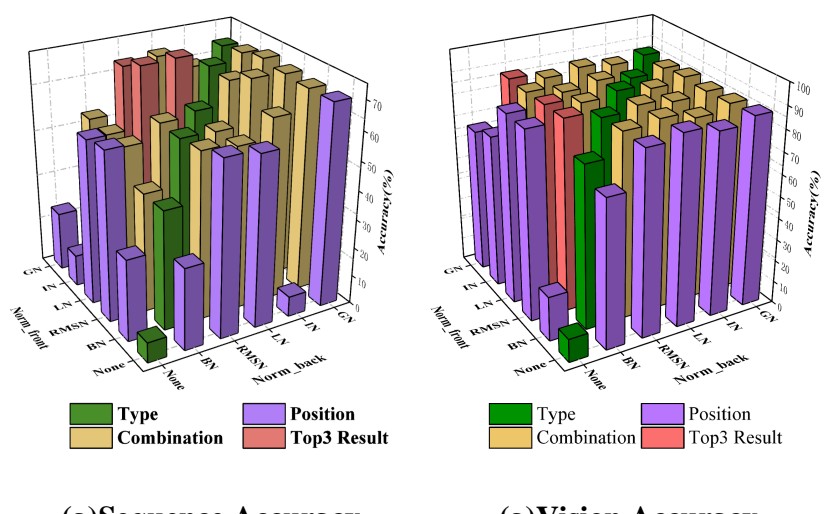

**(a)Sequence Accuracy**          **(a)Vision Accuracy**

Figure 3: The performance bar charts of different normalization combinations in sequence modeling and image classification tasks.

suggesting that after-SSM with BN is particularly effective. **GN**, when used before and after SSM, also maintained high performance (86.3%), consistent with its strong showing in other experiments.

These findings demonstrate that **combining specific different normalization methods before and after SSM** can enhance performance in both tasks. However, the optimal combinations differ between tasks, and no single combination is best for both. **GN before SSM and LN after SSM** continues to perform relative well in various combinations across tasks.

**RECOMMENDATIONS**: The experimental results highlight the critical role of normalization techniques and their positions in neural network architectures. Applying normalization after the SSM module is generally more beneficial in both sequence modeling and image classification tasks. Moreover, combining specific different normalization methods before and after SSM can significantly enhance model performance. **LN** emerges as a versatile and consistently strong performer across tasks, making it a valuable choice for achieving balanced performance. Future work could explore adaptive normalization strategies that dynamically adjust based on task requirements and data characteristics.

## 4.5 VALIDATION EXPERIMENT

To validate our proposal, we evaluated our recommended approach on other datasets and compared it with the original model. For sequence modeling and vision tasks, we conducted experiments on the LRA ListOps dataset and the ImageNet-1k dataset, respectively. The experimental results are shown in Table 5.

Table 5: Performance Comparison Between the Original Model and Our Proposed Approach.

|  | Normalization | Sequence(%) | Normalization | Vision (%) |
|---|---|---|---|---|
| Original | RMSN→SSM→RMSN | 56.9 | LN→SSM→LN | 70.8 |
| Ours | IN→SSM→LN | **72.5** | RMSN→SSM→BN | **71.1** |

For vision tasks, RMSN→SSM→RMSN represents the original Mamba's normalization configuration, while IN→SSM→IN represents our proposed normalization configuration. For vision tasks, LN→SSM→LN represents the original VMamba's normalization configuration without FFN module for fair comparison, while RMSN→SSM→BN represents our proposed normalization configuration. As shown in the table, the experimental results of our proposed approach outperform those of the original model, which verifies the effectiveness of our proposed solution.

## 4.6 INTUITIVE EXPLANATION

We made an intuitive inference that the combination of two specific normalization techniques outperforms normalization alone, but this is not intended as an essential explanation. It is hoped that it will provide insight into understanding this phenomenon.

We conducted an in-depth study on the L2 norms of each layer of Mamba Blocks in a network structure containing four layers of Mamba Blocks on the ListOps dataset from the LRA benchmark, comparing the effects of the same and different normalization methods and plotted the L2 norms of the weights in each Mamba block and conducted an in-depth analysis of the impact of different normalization techniques in a 4-layer model in the following Figure 4.

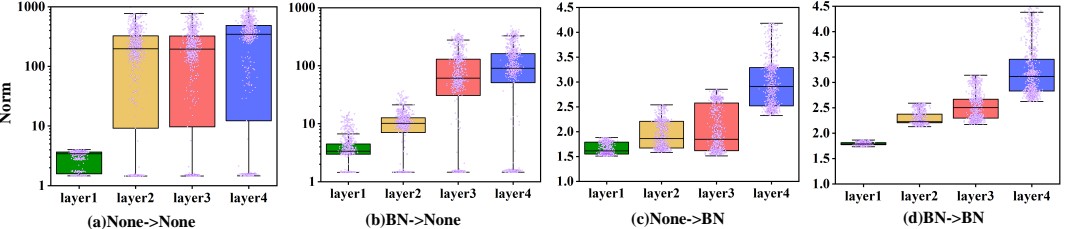

Figure 4: The L2 norm about None→SSM→None, BN→SSM→None, None→SSM→BN, and BN→SSM→BN on the ListOps dataset.

In Figure4, the purple points represent the L2 norm distribution of the current Mamba layer at each iteration step, indirectly reflecting the variation of weights during training. The addition of extra normalization has a significant impact on the L2 norm distribution of weights across different Mamba layers. We observed the following two aspects:

(1) For None→None and BN→None, the L2 norms of weights in different Mamba layers vary significantly. This indicates that the weight magnitudes in deeper layers are substantially larger than those in earlier layers. Additionally, the weight norms in deeper Mamba layers exhibit a highly

polarized distribution, suggesting that the training process may encounter pathological curvature landscapes, leading to training failures.

(2) For None→BN and BN→BN, the methods maintain scale invariance Shleifer et al. (2022) in the weight distribution across all Mamba layers. The L2 norm variations are nearly uniformly distributed across layers, indicating a favorable training landscape. This demonstrates that applying appropriate normalization after the SSM operation helps achieve centered activations and stabilizes gradient updates (LeCun et al., 1990), contributing to robust training dynamics.

Additionally, we tried different normalization combinations and listed the excelled ones in the table4, finding that the optimal results are achieved when different normalization methods are applied before and after the SSM block. However, how can we determine which combinations are the best? Thus, we provide valuable guidance for selecting optimal normalization combinations. For instance, here is the behavior of the combination of BN and IN in Figure5. Our analysis reveals that these two normalization techniques exhibit complementary effects on model weights in the final block, which we call "harmonic structure." We observed that the weight matrix Norm updates in different directions and has a large margin when the two normalizations act alone. The Norm of BN→SSM→IN, on the other hand, is exactly the balance of the two. This leads to a 10% improvement in performance compared to using either normalization individually (see5).

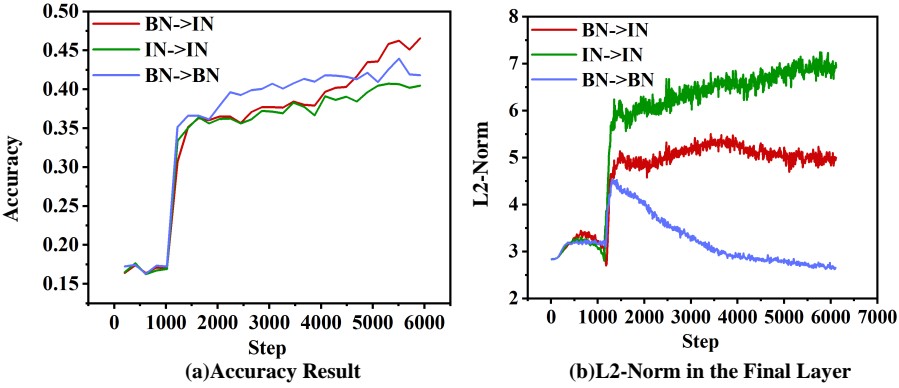

(a)Accuracy Result          (b)L2-Norm in the Final Layer

Figure 5: The Performance and L2 in the fourth layer of BN→SSM→IN on ListOps.

## 5 CONCLUSIONS AND FUTURE WORK

In this study, we conducted a comprehensive investigation into the types, positions, and combinations of normalization layers within the Mamba architecture. Our findings reveal that applying proper normalization after the SSM Modules enhances training stability by mitigating large variations in weight norms. Moreover, specific combinations of normalization techniques not only stabilize the training process but also lead to significant improvements in model performance.

We also propose an intuition for selecting and combining normalization layers to facilitate the exploration of stable training for Mamba and other deep architectures. This intuition provides valuable insights into choosing appropriate normalization methods for robust training of large-scale neural networks. Future research will focus on extending this intuition to more complex models and tasks, aiming to refine normalization strategies for further improvements in efficiency and robustness.

The Mamba2 architecture has recently been introduced, and Dao & Gu (2024) found that training Mamba2 is less stable than Mamba1. Building upon our research into the normalization of Mamba Blocks, future work could further explore the Mamba2 architecture to address its stability challenges. Additionally, we hope our findings can be extended to other deep learning architectures, offering valuable guidance for designing more complex models.

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

## A  NORMALIZATION TAXONOMY

An unified taxonomy was proposed by Huang et al. (2023) to understand the similarities and differences among these methods, specifically including Normalization Representation Area Partitioning (NAP), Normalization Operation (NOP), and Normalization Representation Recovery (NRR). The operational details of these mainstream techniques are clearly presented in the following Table 6.

Table 6: Details of mainstream normalization techniques.

| Method | NAP | NOP | NRR |
|--------|-----|-----|-----|
| BN | $\Pi_{BN}(\mathbf{X}) \in \mathbb{R}^{D \times mL}$ | Standardizing | Learnable $\gamma, \beta \in \mathbb{R}^D$ |
| IN | $\Pi_{IN}(\mathbf{X}) \in \mathbb{R}^{mD \times L}$ | Standardizing | Learnable $\gamma, \beta \in \mathbb{R}^D$ |
| GN | $\Pi_{GN}(\mathbf{X}) \in \mathbb{R}^{mg_D \times s_D L}, g_D \times s_D = D$ | Standardizing | Learnable $\gamma, \beta \in \mathbb{R}^D$ |
| LN | $\Pi_{LN}(\mathbf{X}) \in \mathbb{R}^{m \times DL}$ | Standardizing | Learnable $\gamma, \beta \in \mathbb{R}^D$ |
| RMSN | $\Pi_{RMSN}(\mathbf{X}) \in \mathbb{R}^{m \times DL}$ | Scaling | Learnable $\gamma \in \mathbb{R}^D$ |

Taking an batch sequences $\mathbf{X} \in \mathbb{R}^{m \times D \times T}$ as an example. The NAP operation determines how $\mathbf{X}$ is reshaped into $\mathbf{X} \in \mathbb{R}^{S_1 \times S_2}$, where $S_2$ indexes the sample set used to compute the statistics. For example, in $\Pi_{BN}(\mathbf{X}) \in \mathbb{R}^{D \times (mT)}$, the $mT$ indicates that the statistics (mean and variance) are computed along the batch and sequence length (time steps) dimensions.

## B  TRAINING STABILITY

Given a normed vector space $V$, we refer to the $L_1$ and $L_2$ norm to be the special cases of the following general $L_p$ norm of a give vector $x \in V$, by setting $p = 1$ and $p = 2$:

$$\|x\|_{L^p} = \left( \sum_{j=1}^{\infty} |\xi_j|^p \right)^{1/p}$$

We evaluate the training stability using $L_2$ norms. We analyzed L2 norm of the weight matrix of the whole Mamba Block,including in_projection layer,conv1d layer,SSM Module,out_projection layer.

## C  MAMBA ARCHITECTURE

State space modelling is a method of describing and analysing a system based on matrix theory. Introducing state variables can get more in-depth information about the system. SSM uses first-order differential equations to map the input function $x_t$ to the output function $y_t$ through hidden state $h_t$, defined as follows:

$$h_t = \boldsymbol{A}h_{t-1} + \boldsymbol{B}x_t, \quad y_t = \boldsymbol{C}h_t + \boldsymbol{D}x_t \tag{11}$$

where state transition matrix $\boldsymbol{A} \in \mathbb{R}^{N \times N}$,input matrix $\boldsymbol{B} \in \mathbb{R}^{N \times C}$,output matrix $\boldsymbol{C} \in \mathbb{R}^{C \times N}$ and forward channel transition matrix $\boldsymbol{D} \in \mathbb{R}^{C \times C}$. The variables N and C refer to the hidden state and dimension factors, respectively. Continuous parameters $\boldsymbol{A}, \boldsymbol{B}$ can be discretized by a first-order difference method or a bilinear transformation method as follows $\bar{\boldsymbol{A}}, \bar{\boldsymbol{B}}$, with the sampling interval $\Delta$. Details of the process can be found in (Gu et al., 2021). We give the discretization result directly here. What should be noted is that the hidden state update mechanism of SSM is similar to Recurrent Neural Network (RNN), which receives current time step input $x_t$ and the previous time step hidden state $h_{t-1}$ and computes the current time step hidden state $h_t$.

The key design principle of Mamba lies in the introduction of a selective mechanism to parameterize the transition matrices $\boldsymbol{A}, \boldsymbol{B}, \boldsymbol{C}, \Delta$ in a data-driven manner. In Mamba1, these matrices are defined as functions of the input embedding features, allowing them to adapt dynamically to the data context through a hardware-aware parallel computing algorithm. This enables efficient processing of long sequences with improved computational throughput. In contrast, Mamba2 leverages semi-differentiable matrix factorization to compute the hidden state space efficiently, ensuring that any state space model with state size $N$ and sequence length $L$ can be computed in time $O(TN)$, which means that Mamba2 preserves the controllability of the state space across various sequence lengths.

## D  IMPLEMENTATION DETAILS

### D.1  LISTO EXPERIMENT

**Dataset**  The Long Range Arena (LRA) benchmark (Tay et al., 2021) is designed to evaluate the ability of models to capture long-range dependencies across various tasks. One of the tasks included in the LRA benchmark is the *ListOps* task, which is particularly useful for assessing a model's capability to handle hierarchical structures and process long-range sequences.

In the *ListOps* task, the input sequence is represented as a nested list of operations. These symbolic expressions involve various operations, such as max, min, and median, applied to integers. The task requires the model to parse the input sequence, resolve the nested operations, and compute the final result. For example, a typical input could be:

$$\mathtt{max}(2, \mathtt{min}(3, 9), \mathtt{max}(4, 5))$$

In this example, the operations max, min, and max are applied in a hierarchical structure. The model's objective is to correctly parse the operations and compute the final result, which in this case is 5.

The standard *ListOps* task involves input sequences of length 1000. However, to further test Mamba's capacity to handle long-range dependencies, we used an enhanced dataset, *ListOps-New*, which includes input sequences of length 2000. This extended version allows us to assess better the limits of Mamba's ability to manage and process extremely long sequences with complex, hierarchical operations.

**Experiment Details**  All experiments were implemented in Pytorch(Listo (Tay et al., 2021)) and conducted on 12 NVIDIA 4090 24GB GPU using DDP (Li et al., 2020). We set the initial learning rate as 0.0001 and used the ADAMW optimizer (Kingma & Ba, 2015) for model optimization. The batch size was set to 32. To maintain stable training, we take the cosine_warmup scheduler (Loshchilov & Hutter, 2017). Detailed model configuration information is presented in Table 7

Table 7: Model Configuration Details

| Encoder | Value |
|---|---|
| encoder_name | position |
| encoder_dropout | 0.0 |
| **Layer** | **Value** |
| layer_name | mamba |
| causal | false |
| **Parameter** | **Value** |
| dropout | 0.0 |
| n_layers | 4 |
| d_model | 128 |
| ss_state | 64 |
| d_conv | 4 |
| expand | 2 |
| epoch | 30 |

**Complete Result of Combanation**  We present the results of all possible combinations of commonly used normalization techniques in Table 8. We observed that combinations involving BN and GN often yield better performance. The combinations with GN tend to outperform those with BN, which we hypothesize is due to the inconsistencies between training and inference in BN, leading to performance degradation. The specific reasons behind this require further investigation in future studies.

### D.2  IMAGENET EXPERIMENT

**Experiment Details**  We implemented our solution using VMamba's open-source code (Liu et al., 2024). In the original VMamba VSS Block, it not only includes the Mamba Block but also adds FFN and LN modules afterward. To avoid the impact of these modules and ensure a fair comparison, we

Table 8: Accuracy Comparison for Various Normalization Combinations in listops experiment:

| Normalization | Accuracy% | Normalization | Accuracy% | Normalization | Accuracy% | Normalization | Accuracy% |
|---|---|---|---|---|---|---|---|
| BN → BN | 43.9 | IN → IN | 40.6 | BN → IN | 49.0 | IN → BN | 37.6 |
| BN → BN | 43.9 | RMSN → RMSN | 40.1 | BN → RMSN | 40.1 | RMSN → BN | 44.8 |
| BN → BN | 43.9 | LN → LN | 38.1 | BN → LN | 38.6 | LN → BN | 41.8 |
| BN → BN | 43.9 | GN → GN | 41.4 | BN → GN | 40.1 | GN → BN | 42.9 |
| IN → IN | 40.6 | LN → LN | 38.1 | IN → LN | 39.8 | LN → IN | 41.2 |
| IN → IN | 40.6 | RMSN → RMSN | 40.1 | IN → RMSN | 39.2 | RMSN → IN | 37.2 |
| IN → IN | 40.6 | GN → GN | 41.4 | IN → GN | 46.8 | GN → IN | 44.8 |
| LN → LN | 38.1 | RMSN → RMSN | 40.1 | LN → RMSN | 39.3 | RMSN → LN | 40.0 |
| LN → LN | 38.1 | GN → GN | 41.4 | LN → GN | 40.4 | GN → LN | 42.5 |
| RMSN → RMSN | 40.1 | GN → GN | 41.4 | RMSN → GN | 42.3 | GN → RMSN | 41.6 |

removed the FFN and LN modules in our experiments, while keeping other parameter settings consistent with VMamba's default configuration, such as learning rate, model depth, etc. In subsection 4.5 Validation Experiment, due to computational cost and time constraints, the number of training epochs on the ImageNet-1k dataset was reduced from 300 to 100. The comparative values presented in Table 5 are the results from experiments trained for only 100 epochs.

**Result Showcase**   In Figure 6, we present the training loss and validation accuracy curves during the training of the VMamba model using different normalization layers and positioning them in different locations.

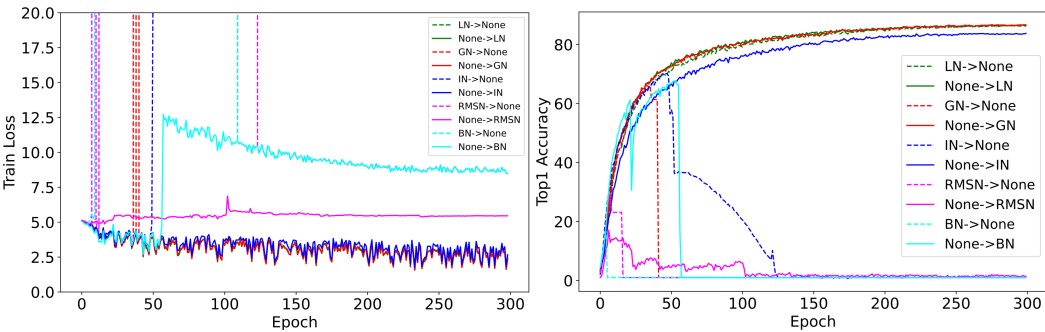

Figure 6: Impact of using five typical Normalization(BN, LN, IN, GN, RMS) in before or after SSM module. The solid line represents the results where normalization is applied after the SSM Module, while the dashed line represents the results where normalization is applied before the SSM Module.

## D.3   MORE EXPERIMENT RESULTS

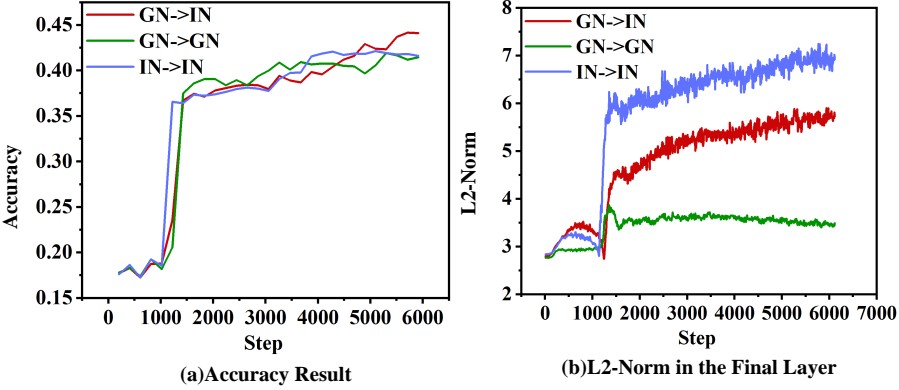

(a)Accuracy Result

(b)L2-Norm in the Final Layer

Figure 7: The Performance and L2 in the last layer of GN→SSM→IN in ListOps(LRA benchmark).

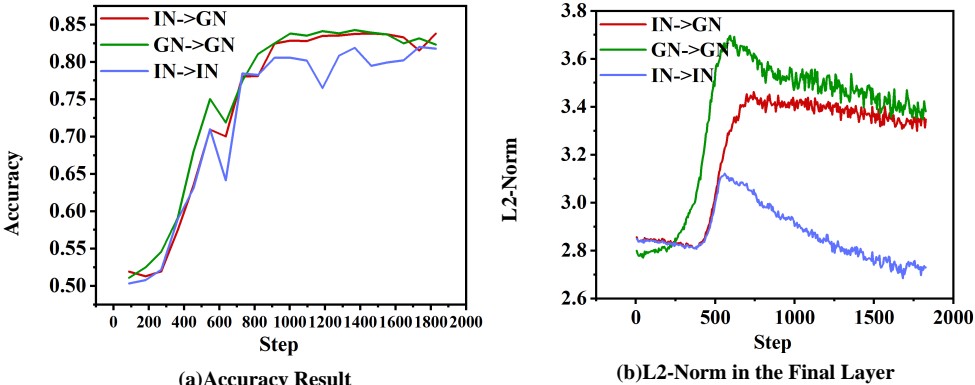

(a)Accuracy Result         (b)L2-Norm in the Final Layer

Figure 8: The Performance and L2 in the last layer of IN→SSM→GN in IMDB(LRA benchmark).

