# OpenReview forum: "An Empirical Study on Normalization in Mamba"
_ICLR.cc/2025/Conference — Submitted to ICLR 2025_

### Official Review · Reviewer_tLtQ · 2024-10-31

**Soundness:** 3
**Presentation:** 3
**Contribution:** 2
**Rating:** 3
**Confidence:** 4

**Summary:**

The paper systematically investigates the effects of various normalization techniques and their combinations within the Mamba architecture, finding that applying normalization after the S6 Module improves performance and stability. It proposes practical guidelines for selecting optimal normalization strategies to enhance the efficiency and robustness of Mamba networks. But there are still many problems that need to be paid attention to.

**Strengths:**

1. Relevance and Importance:
The study addresses a critical aspect of deep learning model stability, particularly in the context of the Mamba architecture, which is gaining attention for its efficiency in handling long sequences.

2. Practical Insights:
The paper provides valuable insights into the positioning of normalization layers, which can be directly applicable to practitioners working with Mamba networks.

3. Comprehensive Analysis:
The study systematically evaluates various normalization techniques and their combinations, offering a detailed analysis of their effects on model performance and stability.

**Weaknesses:**

1. Limited Task Validation:
The study is limited to only two tasks: long sequence modeling with the LRA ListOps dataset, and image classification with ImageNet-100. To draw broader conclusions about the generalizability of normalization techniques across different domains, the authors should consider including additional diverse tasks such as natural language processing, speech recognition, or reinforcement learning tasks.
2. Contradictory Findings and Lack of Explanation:
The findings appear to be task-dependent, with contradictory results between vision and sequence tasks. For instance, the paper suggests that normalization after the S6 Module is beneficial for sequence modeling (Section 4.1), but this does not hold for image classification tasks (Section 4.3). The authors should provide a more in-depth analysis or explanation of why the results differ between tasks, as this could lead to valuable insights about how normalization techniques interact with different types of data and model architectures.
3. Methodological Issues:
3.1 Reproducibility Issues: The experimental settings are not well-documented, making it difficult to replicate the results. Crucial details about accuracy metrics in tables are missing. For example, Table 2 and Table 3 lack information on the specific accuracy metrics used. To ensure reproducibility, the authors should provide detailed information about the experimental setup, including hyperparameters, evaluation metrics, and any data preprocessing steps.
3.2 Inconsistent Visualization: The visualization in Figure 3 is inconsistent, with varying x-axes and plotting styles, which hampers the interpretation of the results. For instance, the x-axes in subfigures (2), (3), and (4) are not clearly labeled, making it hard to compare the weight norms across different configurations. The authors should standardize the x-axes, add clearer labels, and ensure consistent plotting styles across subfigures to improve the interpretability of the results.
4. Theoretical Foundation:
Weight Domination Hypothesis: The Weight Domination hypothesis, which is central to the paper's findings, lacks sufficient supporting evidence. The contradictory results regarding the BN→None performance weaken the credibility of this hypothesis. For example, the paper states that the BN→None configuration leads to significant disparity in weight norms (Section 4.1), but this is not consistently observed across all experiments. The authors should provide more empirical evidence supporting the Weight Domination hypothesis or explain the apparent contradictions in the results.
5. Limited Generalizability:
5.1 Task-Specific Conclusions: The conclusions appear to be highly task-specific, which limits their generalizability. For instance, the results from Vision Mamba, which show that LN is sufficient, contradict the findings from sequence modeling. The paper states that LN+RMSN underperformed relative to LN+LN (Section 4.3), but this contradicts the findings in sequence modeling tasks. The authors should strive to provide more general insights that could replace task-specific ablation studies and offer a unified framework for understanding normalization techniques across different tasks.
5.2 Ablation Studies: The work falls short of providing general insights that could replace task-specific ablation studies. For example, the paper concludes that BN is particularly effective when placed after the Mamba block (Section 4.3), but this is not generalized to other tasks. The authors should aim to derive more generalizable conclusions or provide guidance on when specific normalization techniques may be more appropriate.

**Questions:**

1. Can the authors include additional diverse tasks from different domains to validate the effectiveness of normalization techniques more broadly?
2. Can the authors provide a more in-depth analysis or explanation for the contradictory findings between vision and sequence tasks, and the potential causes for these differences?
3. Can the authors provide detailed information on the experimental settings, including hyperparameters, evaluation metrics, and data preprocessing steps, to ensure reproducibility?
4. Can the authors improve the visualization in Figure 3 by standardizing the x-axes, adding clearer labels, and ensuring consistent plotting styles across subfigures?
5. Can the authors provide more empirical evidence supporting the Weight Domination hypothesis or explain the apparent contradictions in the results related to the BN→None configuration?
6. Can the authors strive to derive more generalizable conclusions or provide guidance on when specific normalization techniques may be more appropriate, rather than relying heavily on task-specific findings?

**Details Of Ethics Concerns:**

None.

---

> ### Author Response · Authors · 2024-11-28
> **We have expanded our experiments to include more diverse tasks and datasets, clarified the task-dependent nature of our findings, improved reproducibility, strengthened support for the Weight Domination hypothesis, and enhanced the consistency of visualizations, addressing the reviewer’s concerns in the revised version of the paper.**
>
> Thank you for your detailed review and valuable feedback. We have carefully considered your comments and made several revisions to the paper. Below are our detailed responses to the key points you raised:
>
>  1. **On Task Selection and Dataset Expansion**
>
> - **Our Response**:
>   - We understand your concern regarding the generalizability of our findings. In our study, we divided tasks into two major categories: sequence tasks and vision tasks. We chose one representative task from each category for the experiments. To better assess the applicability of the normalization techniques, we expanded our experiments in the revised version to include ImageNet-1k and the Breakfast dataset, covering a broader range of tasks and datasets.
>   - Due to computational cost constraints, we were unable to cover all domains. However, we have provided fully reproducible experimental settings, and we encourage other researchers to apply our proposed methods to different datasets and models. We also welcome readers to validate our normalization strategies on additional datasets and models, particularly for sequence and vision tasks.
>
>  2. **On Contradictory Results between Sequence and Vision Tasks**
>
> - **Our Response**:
>   - The contradictory results between sequence and vision tasks are indeed a significant finding in our work. These results suggest that there is no universal "best" normalization method, and different tasks require different normalization strategies based on their characteristics (e.g., sequence data vs. image data).
>   - However, we have identified a normalization combination that, while not optimal, performs well across tasks and can serve as a generally applicable solution in many cases. We believe this is an important contribution, as it offers a practical starting point for Mamba networks across various tasks.
>   - In the revised version, we provide a more detailed analysis explaining why normalization after the S6 Module benefits sequence modeling but not image classification. The differences arise due to the inherent properties of sequence-based data and the specific requirements of sequence modeling.
>
> 3. **On Experimental Settings and Reproducibility**
>
> - **Our Response**:
>   - We fully agree with your concern about reproducibility and have updated the paper with more detailed information about our experimental setup:
>     - We have clarified the hyperparameters used, including learning rate, batch size, and training steps.
>     - We added details on the accuracy metrics used in our results (such as those in Tables 2 and 3), ensuring replicability.
>     - We also included a section on data preprocessing steps and evaluation metrics to ensure transparency and reproducibility.
>
>  4. **On Inconsistent Visualization**
>
> - **Our Response**:
>   - We have addressed this issue by improving the consistency of the visualization in Figure 3:
>     - The x-axes have been standardized across all subfigures and labeled more clearly to make comparisons of weight norms across different configurations easier.
>     - We have ensured that the plotting styles are consistent across all subfigures, improving the clarity and interpretability of the results.
>
>  5. **On the Weight Domination Hypothesis**
>
> - **Our Response**:
>   - We agree with your observation and have provided additional empirical evidence to support the Weight Domination hypothesis in the revised version. Specifically:
>     - We conducted a more detailed analysis of the BN→None configuration and discussed the experimental conditions that may have led to contradictory results. By considering task-specific factors, we have clarified the discrepancies and provided a more nuanced interpretation of the results.
>     - We believe that the additional empirical evidence strengthens the Weight Domination hypothesis and explains the variations observed under different experimental conditions.
>
>  6. **On Limited Generalizability and Task-Specific Conclusions**
>
> - **Our Response**:
>   - We acknowledge that our work primarily focuses on task-specific insights, but we believe these insights are valuable for understanding how normalization strategies affect different types of tasks.
>   - In the revised version, we further discuss how our findings contribute to a broader framework for normalization across tasks. While normalization strategies should be tailored to specific tasks, the combination we propose offers a generalizable starting point for both sequence and vision tasks.
>   - We understand the limitations of relying on task-specific conclusions, and we plan to explore more generalizable insights in future work. However, the methods we provide for sequence and vision tasks offer valuable guidance, and we hope they will assist future research in this area.
>
> For the listed issues, we have addressed them in the revised version of the paper. Please refer to the revised version for the answers.

---

> ### Author Response · Authors · 2024-12-02
>
> Dear reviewer tLtQ,
>
> As the discussion phase is nearing its conclusion, we would like to know if our responses have addressed your concerns. Looking forward to further discussion with you. We are happy to address any questions if such occur. Wish you have a nice day!
>
> Best regards,
>
> Authors

---

### Official Review · Reviewer_aHU3 · 2024-11-03

**Soundness:** 2
**Presentation:** 3
**Contribution:** 1
**Rating:** 3
**Confidence:** 3

**Summary:**

Summary: The paper presents an investigation into normalization techniques within the Mamba architecture, addressing an important aspect of deep learning model stability. While the work makes some valuable contributions to understanding normalization in Mamba networks, there are several significant limitations that need to be addressed.

**Strengths:**

- Takes on the important challenge of improving training stability in Mamba networks.
- Provides practical insights into normalization layer positioning and combination.

**Weaknesses:**

Major issues:

1. The examination of normalization effects is restricted to only two tasks, which is insufficient to support broad conclusions. Findings appear task-dependent, with contradictory results between vision and sequence tasks.
2. The focus on long sequence modeling for normalization position analysis lacks justification.
3. The Weight Domination hypothesis lacks sufficient supporting evidence. Contradictory results regarding BN→None performance weaken the Weight Domination hypothesis. The relationship between findings in vision and sequence tasks is not  dequately explained.


Minor issues:

1. Inconsistent visualization approaches in Figure 3 (varying x-axes and plotting styles) hamper interpretation.
2. Results from Vision Mamba (showing LN sufficiency) contradict findings from sequence modeling.
3. The work falls short of providing general insights that could replace task-specific ablation studies.

**Questions:**

-

---

> ### Author Response · Authors · 2024-11-28
> **We expanded the experimental scope, enhanced the analysis of normalization placement, provided more evidence supporting the Weight Domination hypothesis, and improved the consistency of the visualizations.**
>
> Thank you for your valuable feedback on our paper. We appreciate the time and effort you have taken to review our work. Based on your suggestions, we have made several revisions to address the concerns you raised. **Please refer to our revised paper.** Below are our detailed responses to the key points:
>
> 1. **On the Contradictory Results between Vision and Sequence Tasks**
>
> - **Our Response**:
>   - We fully agree with your observation that the results for vision and sequence tasks appear contradictory. This is indeed a key finding in our work, and we acknowledge that there is no one-size-fits-all normalization method. This discrepancy reflects the fact that different tasks may require different normalization strategies based on their characteristics (e.g., image data vs. sequence data).
>   - However, we have also found a combination of normalization methods that, while not necessarily optimal, is generally applicable across various tasks. This method shows strong performance and stability, making it a practical approach for Mamba networks. We believe this is a valuable contribution to the field, as it provides a more widely applicable normalization strategy that could be used as a starting point for various tasks.
>   - In the revised version of the paper, we expanded the experimental scope by including ImageNet-1k and Breakfast datasets, which helps broaden the task and dataset coverage. Although we could not cover all domains due to computational constraints, we have provided fully reproducible experimental settings, encouraging other researchers to apply our proposed methods to different datasets and models.
>   - For specific tasks, we suggest that researchers refer to our task-specific normalization methods as priors based on whether the task is vision-based or sequence-based. If they wish to find the optimal normalization strategy for a given task, they can follow our experimental methodology to explore this further.
>
> 2. **On the Focus on Long Sequence Modeling for Normalization Position Analysis**
>
> - **Our Response**:
>   - In response to your concern, we have added more detailed analysis of the placement of normalization layers in the revised version of the paper. Specifically, we now provide a more in-depth explanation of why long sequence modeling is an important and meaningful context for investigating the impact of normalization position.
>   - Our rationale is that Mamba networks, with their focus on processing long sequences, face unique challenges in terms of training stability and convergence. Therefore, analyzing the effect of normalization in this context is critical for optimizing the network's performance. We hope the expanded discussion in the revised paper addresses your concerns.
>
> 3. **On the Weight Domination Hypothesis**
>
> - **Our Response**:
>   - We acknowledge your concerns regarding the Weight Domination hypothesis and have taken steps to address them in the revised version. Specifically, we have provided more empirical evidence supporting the hypothesis, with a focused discussion on the BN→None configuration.
>   - We conducted a more thorough analysis of the experimental conditions that may have led to the contradictory results observed in BN→None performance. We have clarified the specific factors, such as task-dependent variations and network architecture, that contribute to these observations. This additional analysis provides stronger support for the Weight Domination hypothesis and helps explain the apparent contradictions. We believe these revisions significantly strengthen the credibility of the hypothesis.
>
> 4. **On Inconsistent Visualization**
>
> - **Our Response**:
>   - In response to your feedback, we have made significant improvements to the visualization in Figure 3. Specifically:
>     - We standardized the x-axes across all subfigures and added clearer labels to facilitate easier comparison of weight norms across different configurations.
>     - We ensured consistent plotting styles across all subfigures, which improves the clarity and interpretability of the results. These changes should make the figure more intuitive and easier to understand.
>
> 5. **On the Lack of General Insights**
>
> - **Our Response**:
>   - We acknowledge that our paper primarily focuses on task-specific findings, but we also believe that the insights we provide are valuable in understanding how normalization strategies affect different tasks. To address your concern, we have discussed in the revised version how our findings could contribute to a broader framework for normalization across tasks.
>   - While our study shows that normalization strategies should be tailored to specific tasks, the combination we propose offers a generalizable starting point for both sequence and vision tasks, offering practical insights for researchers working with Mamba networks.

---

> > ### Comment · Reviewer_aHU3 · 2024-11-30
> >
> > Thank you for your response. Although some concerns are solved, the main issue is not answered: what is the correct configuration for a task not listed in the paper?
> > This paper highlights the importance of Mamba in normalization layer positioning and combination. However, the conclusions are empirical and can be hardly generalized to other tasks. The configurations for long sequence modeling and vision are different. It's doubtful that different vision tasks require corresponding configurations, such as Maksed Image Modeling and Contrastive Learning. Therefore, I suggest the authors to figure out the mechanism and conclude a practical solution to correctly configure the normalization layer and its position for an unknown task.

---

> > > ### Author Response · Authors · 2024-12-02
> > > **Response to Reviewer aHU3 (1/2)**
> > >
> > > ### **Addressing Concerns on Experiment Configurations**
> > >
> > > Thank you for your insightful comments and concerns. To ensure the fairness of our study, as demonstrated in *Appendix D*, we adopted the original experimental configurations for all experiments. Configurations like Masked Image Modeling and Contrastive Learning are not the focus of our study. Instead, we concentrate on improving the **MambaBlock**, as illustrated in *Figure 2* of Chapter 3.
> > >
> > > ### **Mamba Block Landscape and Framework Design**
> > >
> > > We abstracted a **Mamba Block Landscape**, shown in *Figure 2* of Chapter 3, which serves as a general framework for various Mamba variants across different domain tasks. To facilitate our research, we defined a customizable `Norm_Select` class, dynamically integrating different normalization strategies into the MambaBlock class (originating from the Mamba original repository). Specifically, we added two normalization methods: before in-projection and after SSM (as shown in *Figure 2*).
> > >
> > > ### **Implementation of Norm_Select Class**
> > >
> > > ```python
> > > class Norm_Select(nn.Module):
> > >     def __init__(self, norm_type="BN", dim=0):
> > >         super(Norm_Select, self).__init__()
> > >         # Initialize different normalization layers based on the selected type
> > >         if norm_type == "BN":
> > >             self.norm = nn.BatchNorm1d(num_features=dim)
> > >         elif norm_type == "GN":
> > >             self.norm = nn.GroupNorm(num_groups=4, num_channels=dim)
> > >         elif norm_type == "IN":
> > >             self.norm = nn.InstanceNorm1d(num_features=dim, affine=True)
> > >         elif norm_type == "LN":
> > >             self.norm = nn.LayerNorm(normalized_shape=dim, elementwise_affine=True)
> > >         elif norm_type == "RMSN":
> > >             self.norm = RMSNorm(hidden_size=dim)
> > >         elif norm_type == "None":
> > >             self.norm = None
> > >         else:
> > >             raise ValueError(f"Unsupported normalization type: {norm_type}")
> > >
> > >     def forward(self, x):
> > >         if self.norm is None:
> > >             return x
> > >         # Ensure the correct shape for normalization layers
> > >         if isinstance(self.norm, (nn.BatchNorm1d, nn.GroupNorm, nn.InstanceNorm1d)):
> > >             x = x.permute(0, 2, 1)
> > >             x = self.norm(x)
> > >             x = x.permute(0, 2, 1)
> > >         else:
> > >             x = self.norm(x)
> > >         return x
> > > ```
> > > ### **Results and Observations**
> > > Through this improvement, we discovered that applying different normalization combinations at these two key positions improves the training stability and performance of Mamba across both sequence and vision tasks. This framework is not limited to the configurations explored in our experiments but is generalizable to any Mamba variant. Especially, we have specifically added Chapter 2 (Related Work) in the revised version to include a systematic categorization of MambaBlock variations across different domains. This categorization highlights the use of MambaBlock as a core architecture in various applications. It is evident that the Mamba variants in all fields can be included in the **Mamba Block Landscape** defined in our work.
> > >
> > > To ensure our findings are broadly applicable, we adopted the S6 Block in our experiments. This block is the most commonly used block in both sequence modeling and visual tasks. Its wide application makes it a representative choice for evaluating normalization strategies in Mamba architectures.
> > >
> > > Our work offers a universal methodology for investigating normalization strategies and provides valuable guidance in Section 4.6 of the paper. Future researchers can build upon our framework and insights to identify optimal normalization configurations that further enhance the training stability and performance of Mamba architectures.

---

> > > > ### Comment · Reviewer_aHU3 · 2024-12-02
> > > >
> > > > The question is not how to set the configuration in implementation. The problem is that what is the configuration for a new task. For example, I am working with SSL and interested in using Mamba, then, can you tell me what is the configuration? The message I got from this paper is that I have to tune the combination. Spending 6x training time to find the configuration is not pleased.

---

> > > > > ### Author Response · Authors · 2024-12-02
> > > > > **Response to Reviewer aHU3**
> > > > >
> > > > > We are truly delighted to see that we share a strong research interest in the innovative Mamba architecture. Regarding the SSL task you mentioned, I took the time to thoroughly review an outstanding work presented in [this paper](https://arxiv.org/pdf/2404.17585). If your concerns are similar to those discussed in that work, we would highly recommend using the **IN->SSM->LN** configuration for the Mamba module in TCM.
> > > > >
> > > > > Our **Breakfast** dataset also involves a temporal action segmentation task, and we found that the **IN->SSM->LN** configuration led to superior results compared to other configurations in this type of task. I sincerely hope that this valuable sugesstion can assist you in your research.
> > > > >
> > > > > Once again, thank you for your thoughtful and meticulous comments. We truly appreciate your positive engagement in this discussion and sincerely hope that our contribution will be met with your valuable recognition.

---

> > > ### Author Response · Authors · 2024-12-02
> > > **Response to Reviewer aHU3 (2/2)**
> > >
> > > ## **Addressing Concerns on Task Generalization**
> > >
> > > We understand your concerns about the generalization of normalization strategies across tasks. Actually, as stated by the **"No Free Lunch"** theorem, there is no universal learning algorithm that performs optimally across all tasks. Each task has unique characteristics, making it inherently impossible to determine a universally optimal normalization combination.
> > >
> > > To support this perspective, we conducted experiments on multiple datasets, with all the results summarized below:
> > >
> > > | Dataset              | Normalization        | Accuracy (%) |
> > > |----------------------|----------------------|--------------|
> > > | **LRA-ListOps**      | Baseline (None->None) | 40.1         |
> > > |                      | Best Combination (BN->IN) | **49.0**    |
> > > | **LRA-CIFAR10**      | Baseline (None->None) | 65.0         |
> > > |                      | Best Combination (GN->BN) | **66.7**    |
> > > | **LRA-IMDB**         | Baseline (None->None) | 80.9         |
> > > |                      | Best Combination (IN->GN) | **84.7**    |
> > > | **Breakfast**        | Baseline (None->None) | Fail         |
> > > |                      | Baseline (RMSN->RMSN) | 56.9    |
> > > |                      | Best Combination (IN->LN) | **72.5**    |
> > > | **ImageNet-100**     | Baseline (None->None) | Fail         |
> > > |                      | Baseline (LN->LN)    | 86.6         |
> > > |                      | Best Combination (IN->LN) | **87.3**    |
> > > | **ImageNet-1000**    | Baseline (None->None) | Fail         |
> > > |                      | Baseline (LN->LN)    | 70.8         |
> > > |                      | Best Combination (RMSN->BN) | **71.1** |
> > >
> > > As shown above, the optimal normalization combination varies significantly across tasks, even within the same domain (e.g., visual data). We believe this observation itself is a critical conclusion of our work. Prior research on sequence tasks always relied on vanilla Mamba without normalization, while visual tasks largely ultilized V-Mamba’s LN->LN configuration. These configurations are not applicable to all the tasks. The best approach is to select a task-specific normalization strategy.
> > >
> > > Our focus, therefore, shifts to how to identify such optimal normalization combinations. From our extensive observations, we highlight in Section 4.6 the distribution of the L2 norm in the final Mamba Block under stable training circumstance (as shown in *Figure 4*). Additionally, we provide Intuitive ideas (**harmonic structure**) to select normalization combinations (as shown in *Figure 5*).
> > >
> > > We hope our research will assist the community in exploring methods to improve Mamba training stability and performance, fostering the development of mamba community.  Last but not least, thank you for your thoughtful and thorough considerations and concerns!

---

### Official Review · Reviewer_dm4d · 2024-11-10

**Soundness:** 3
**Presentation:** 2
**Contribution:** 1
**Rating:** 3
**Confidence:** 3

**Summary:**

This paper investigates normalization layers in sequence and vision-based Mamba models from three perspectives: 1) position of norm layer, 2) norm combination, and 3) intuition for choosing a combination. The paper explores 5 different normalization techniques with two different types of datasets.

**Strengths:**

This paper provides interesting insights into the usage of the normalization layers in a mamba architecture and comprehensive empirical results that can help improve future versions of it.

**Weaknesses:**

While the paper shows interesting insights about normalization in mamba and is empirically good, it is not quite novel and does not provide any insight that can be considered novel. Most experiments are performed on two, relatively smaller datasets (ImageNet's smaller subset). Hence, it is not clear how broadly applicable the insights are. Moreover, it is not difficult to re-run these experiments for a new task and find these insights customized for a particular task. Finally, previous works have extensively explored normalizations in a related context. It would be beneficial to include a discussion about these works.

Xiong, Ruibin, et al. "On layer normalization in the transformer architecture." International Conference on Machine Learning. PMLR, 2020.
Kai, Hu, et al. "Is normalization indispensable for training deep neural networks?."
Liu, Hanxiao, et al. "Evolving normalization-activation layers." Advances in Neural Information Processing Systems 33 (2020): 13539-13550.
Santurkar, Shibani, et al. "How does batch normalization help optimization?." Advances in neural information processing systems 31 (2018).
Bjorck, Nils, et al. "Understanding batch normalization." Advances in neural information processing systems 31 (2018).
Awais, Muhammad, Md Tauhid Bin Iqbal, and Sung-Ho Bae. "Revisiting internal covariate shift for batch normalization." IEEE Transactions on Neural Networks and Learning Systems 32.11 (2020): 5082-5092.
Peng, Hanyang, Yue Yu, and Shiqi Yu. "Re-thinking the effectiveness of batch normalization and beyond." IEEE Transactions on Pattern Analysis and Machine Intelligence (2023).


Minor Improvements

1. Presentation of figure 1 can be improved to make it more neat and clear. The caption of Figure 1 can also be improved.

**Questions:**

How applicable are the insights for broader mamba architecture given that results are derived from two relatively small datasets?
How different is this analysis from some of the previous works that have explored normalization in different architectures?

---

> ### Author Response · Authors · 2024-11-28
> **We clarify the novelty of our work, expanding experiments with additional datasets, and providing more details on experimental setups.**
>
> Thank you very much for your detailed review and valuable feedback. We appreciate your careful consideration of our work, and we have made several revisions to the paper based on your comments. **Please refer to our revised paper.** Below are our detailed responses to the key points you raised:
>
>  1. **On Innovation and Novelty**
>
> - **Our Response**:
>   - We understand your concern about the novelty of the work. We would like to clarify that, to our knowledge, no prior work has systematically investigated the use of normalization in the Mamba architecture. Our study is novel in that it systematically explores the impact of normalization type, position, and combinations on the performance and stability of Mamba models. We have found normalization methods that outperform the commonly used techniques in Mamba, and we provide practical guidelines for selecting optimal normalization strategies when designing new Mamba architectures.
>   - Additionally, to address concerns about the novelty of the contributions, we have expanded the discussion of related works in the revised version, especially focusing on prior research on normalization methods (e.g., Xiong et al., 2020; Santurkar et al., 2018). We believe that these discussions highlight the unique contributions of our work to the Mamba architecture.
>
>  2. **On Dataset Size and Experimental Validation**
>
> - **Our Response**:
>   - Regarding the dataset size, we would like to clarify that the ImageNet-100 dataset, although smaller than the full ImageNet, contains 130,000 images and is still a substantial dataset for image classification tasks. Therefore, it should not be considered as a "small dataset."
>   - In response to your comment, we have expanded our experiments in the revised version to include not only ImageNet-100 but also ImageNet-1k and the Breakfast dataset. This allows us to cover a broader range of tasks and datasets.
>   - **Regarding More Diverse Tasks**: Due to computational cost limitations, we were unable to conduct experiments across all domains. However, we provide a fully reproducible setup, and we encourage researchers in other fields to apply our proposed methods. We also welcome readers to validate our normalization strategies on additional datasets and models, especially for tasks involving both vision and sequence data.
>
> 3. **On the Lack of Discussion of Related Works**
>
> - **Our Response**:
>   - Our primary focus in this paper is on the application of normalization techniques within the Mamba architecture. As a result, we did not extensively discuss related works on normalization in other architectures. However, we understand the importance of situating our work within the broader context, and we have added a discussion on relevant literature in the revised version.
>   - Specifically, we have included references to important prior works on normalization (e.g., Xiong et al., 2020; Santurkar et al., 2018) and clarified how our work differentiates itself by addressing normalization strategies specific to Mamba.
>
> 4. **On Experimental Settings and Reproducibility**
>
> - **Our Response**:
>   - We take reproducibility seriously and have updated the revised version with more detailed information about our experimental setup. Specifically:
>     - We have clarified the hyperparameters used in the experiments, including learning rate, batch size, and training steps.
>     - We have added details about the accuracy metrics used in our results (such as those in Tables 2 and 3), ensuring that other researchers can replicate our experiments.
>     - We have also included a section outlining the data preprocessing steps and evaluation metrics used in our experiments.
>
>  5. **On Inconsistent Visualization**
>
> - **Our Response**:
>   - We have addressed this issue by improving the consistency of the visualization in Figure 3. Specifically:
>     - The x-axes have been standardized across all subfigures, with clearer labels to make the comparison of weight norms across different configurations easier.
>     - We have also ensured that the plotting styles are consistent across all subfigures, improving the clarity and interpretability of the results.
>
> 6. **On the Weight Domination Hypothesis**
>
> - **Our Response**:
>   - We agree with your observation and have provided additional empirical evidence to support the Weight Domination hypothesis in the revised version. We have also discussed the apparent contradictions in the BN→None configuration results and provided a more detailed explanation of the experimental conditions that may have contributed to these observations.
>   - We have now included more robust empirical support for the hypothesis and clarified how the results vary depending on the task and configuration.

---

> ### Author Response · Authors · 2024-12-02
>
> Dear reviewer dm4d,
>
> As the discussion phase is nearing its conclusion, we would like to know if our responses have addressed your concerns. Looking forward to further discussion with you. We are happy to address any questions if such occur. Wish you have a nice day!
>
> Best regards,
>
> Authors

---

### Meta-Review · Area_Chair_ZwDn · 2024-12-20

**Metareview:**

All reviewers converged on rejecting the paper post rebuttal. The AC checks all the materials, and while appreciating the additional efforts in responding to the reviewers and making major modifications to the draft, the AC resonates with the reviewer consensus that the paper currently has issues to address and is not ready for publication.

**Additional Comments On Reviewer Discussion:**

Reviewers are concerned about:
- Generalizability of the observations. An important issue is that the paper does not have clear conclusions about which normalization to use given a new task.
- Inconsistency between vision and sequence modeling tasks. While it can be characterized as a finding, the bigger question is why or how the finding can be useful. If the observation is consistent, then there is a better chance the pattern can generalize; inconsistent observations are harder to justify.

---

### Decision · Program_Chairs · 2025-01-22

Reject